# OptPipe: Memory- and Scheduling-Optimized Pipeline Parallelism for LLM Training

## Abstract

Pipeline parallelism (PP) has become a standard technique for scaling large language model (LLM) training across multiple devices. However, despite recent progress in reducing memory consumption through activation offloading, existing approaches remain largely heuristic and coarse-grained, often overlooking the fine-grained trade-offs between memory, computation, and scheduling latency. In this work, we revisit the pipeline scheduling problem from a principled optimization perspective. We observe that prevailing strategies either rely on static rules or aggressively offload activations without fully leveraging the interaction between memory constraints and scheduling efficiency. To address this, we formulate scheduling as a constrained optimization problem that jointly accounts for memory capacity, activation reuse, and pipeline bubble minimization. Solving this model yields fine-grained schedules that reduce pipeline bubbles while adhering to strict memory budgets. Our approach complements existing offloading techniques: whereas prior approaches trade memory for time in a fixed pattern, we dynamically optimize the tradeoff with respect to model structure and hardware configuration. Experimental results demonstrate that our method consistently improves both throughput and memory utilization. In particular, we reduce idle pipeline time by up to 50% under the same per-device memory limit, and in some cases, enable the training of larger models within limited memory budgets. Our code is available[1].

## 1 Introduction

As large language models (LLMs) continue to grow in size and complexity, traditional data parallelism (Goyal et al., 2017) is no longer sufficient, as a single device cannot store the entire model. To address this limitation, model parallelism (Harlap et al., 2018; Huang et al., 2019; Shoeybi et al., 2019; Zheng et al., 2022) partitions the model across multiple devices, making efficient multi-device training a central challenge. Among model parallelism techniques, pipeline parallelism (PP) (Huang et al., 2019; Harlap et al., 2018) is widely adopted: it divides the model into stages, allowing devices to process different segments concurrently. Compared with approaches such as ZeRO (Rajbhandari et al., 2021) and tensor parallelism (Shoeybi et al., 2019), PP generally incurs lower communication overhead. However, PP also introduces new scalability challenges, particularly the trade-off between activation memory consumption and device utilization lost to pipeline bubbles. As the number of pipeline stages increases, the memory required to store intermediate activations can quickly become a bottleneck.

One line of work that improves the PP is to improve the efficiency by reducing the pipeline bubbles. A notable scheduling strategy to address the limitation is *one-forward-one-backward* (1F1B) (Fan et al., 2021; Narayanan et al., 2021), which provides faster memory clearance by early scheduling backward passes. Based on 1F1B, *Interleaved* 1F1B(Fan et al., 2021) further reduces pipeline bubbles while increasing peak memory usage and communication overhead. Then, Zero Bubble (Qi et al., 2023) and Interleaved Zero Bubble (Qi et al., 2024) further improve the efficiency of PP by splitting the backward pass into two parts, backward pass for weight and backward pass for activation, which obtains a zero bubble ratio by flexibly scheduling the backward pass for weight.

---

[1] https://anonymous.4open.science/r/OptPipe-BF38

However, these methods require a large amount of memory to store the activations, which is not suitable for training models with a large number of PP stages.

Activation offloading (Wu et al., 2024; Chen et al., 2025; Wan et al., 2025) represents another line of work aimed at reducing the memory footprint of pipeline parallelism (PP). The key idea is to offload intermediate activations from device memory to host memory, which is typically large enough to store all activations. While this enables training larger models with fewer devices, it also introduces nontrivial scheduling challenges and may increase pipeline bubbles if not carefully managed. PipeOffload (Wan et al., 2025) addresses this issue by selectively offloading activations with long lifespans and low transfer cost, thereby reducing peak memory usage while largely preserving PP efficiency. However, PipeOffload cannot further reduce bubbles through backward-pass splitting due to scheduling complexity, and it relies on simple heuristics for offloading decisions, which can be far from optimal in practice.

In this work, we propose OptPipe, a new pipeline scheduling approach that integrates activation offloading with fine-grained scheduling and backward-pass splitting. We formulate the scheduling problem, with or without activation offloading, as a Mixed-Integer Linear Programming (MILP) model and solve it using both commercial solvers and specialized heuristics designed for this setting. In addition, by parallelizing the solving process, we hide solver overhead and significantly improve the practical efficiency of our method.

Our contributions can be summarized as follows:

- We formulate the pipeline parallelism scheduling problem, both with and without activation offloading, as an MILP model, which yields the optimal scheduling strategy.
- We propose a new PP approach, OptPipe, which integrates specialized heuristics for solving the MILP formulation and additional strategies that enhance its practical implementation.
- We conduct extensive experiments on diverse models and datasets, demonstrating that OptPipe significantly improves training efficiency while maintaining memory usage within device limits.

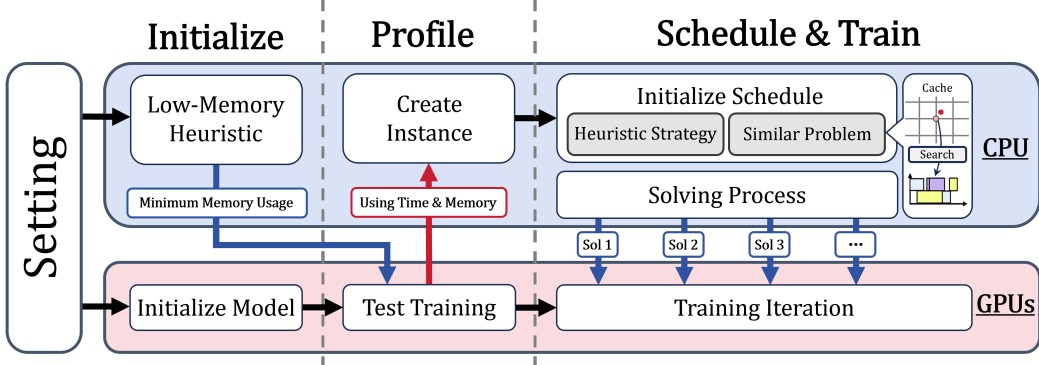

Figure 1: The framework of OptPipe. The framework consists of three main phases: (1) **Initialize**: Generate an initial scheduling strategy that ensures peak memory usage remains within device limits; (2) **Profile**: Run a few warm-up iterations to profile computation time and memory usage, and use the collected statistics to construct the MILP model; (3) **Schedule & Train**: Implement an initial schedule using a heuristic strategy (e.g., PipeOffload or a cached schedule), and employ an MILP solver to refine the schedule. Training proceeds in parallel, with updates applied whenever the solver discovers an improved solution.

## 2 PRELIMINARY

### 2.1 PIPELINE PARALLELISM

Pipeline parallelism (PP) is a form of model parallelism designed to overcome the memory constraints of training large-scale models, particularly those with tens of billions of parameters. When

the size of a model's weights exceeds the aggregate memory of GPUs on a single server, neither data parallelism nor tensor parallelism provides an efficient solution. For example, tensor parallelism often suffers from bottlenecks due to low-bandwidth inter-node communication. PP mitigates this by partitioning a model's layers vertically across multiple GPUs, often spanning several nodes. In this setup, each GPU stores and processes only a subset of the model's layers, thereby reducing the per-device memory footprint.

Formally, consider a model with $N$ layers partitioned into $P$ stages, where each stage contains $N/P$ consecutive layers. In this work, we focus on the case where each stage is assigned continuous layers. For instance, in a four-GPU system with a 32-layer model, GPU 1 may hold layers 1–8, GPU 2 layers 9–16, GPU 3 layers 17–24, and GPU 4 layers 25–32.

## 2.2 COMPUTATION GRAPH IN PIPELINE PARALLELISM

Following the simplified computation model commonly used to analyze efficiency in prior work (Shoeybi et al., 2019; Qi et al., 2023), pipeline parallelism involves scheduling two main components: the forward pass and the backward pass, as illustrated in Figure 2.2.

During the forward pass, the activations from the previous stage, denoted as $x_i$, are passed as input to the next stage (Stage$_{i+1}$). At this stage, the input is first transformed linearly, $z = Wx_i$, where $W$ is the weight matrix of Stage$_{i+1}$. The result $z$ is then processed by a non-linear activation function $\sigma(\cdot)$, producing the stage output $x_{i+1} = \sigma(z)$, which is forwarded to the subsequent stage.

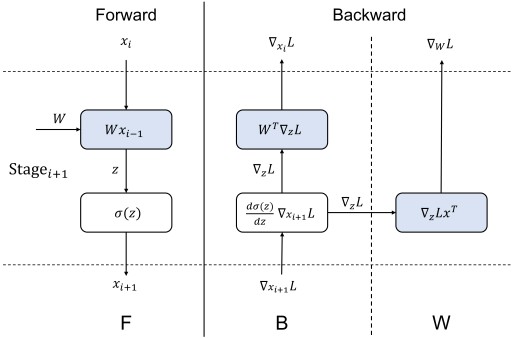

Figure 2: The computation graph of each stage in pipeline parallelism.

The backward pass propagates gradients in the reverse direction. Each stage begins by receiving the gradient of the loss with respect to its output, $\nabla_{x_{i+1}} L$ where $L$ denotes the loss function, from the subsequent stage. Applying the chain rule through the activation function yields the gradient with respect to $z$: $\nabla_z L = \frac{d\sigma(z)}{dz} \nabla_{x_{i+1}} L$. This intermediate gradient serves two purposes. First, to continue backpropagation, the gradient with respect to the stage input is computed as $\nabla_{x_i} L = W^\top \nabla_z L$, and passed to the previous stage. Second, the gradient with respect to the stage weights is obtained as $\nabla_W L = \nabla_z L x_i^\top$, which is subsequently used by the optimizer to update the model parameters.

## 3 SCHEDULING VIA MIXED-INTEGER LINEAR PROGRAMMING

To identify the optimal schedule that minimizes total training time, we formulate the pipeline scheduling problem as an MILP model. The objective of the model is to determine the exact timing of all computational and data-transfer operations, while simultaneously making strategic decisions on whether to offload activation memory for each operation. These decisions are subject to hardware limitations and data-dependency constraints. This section presents a high-level overview of the model, with the complete formulation provided in Appendix C.

### 3.1 MODELING FRAMEWORK

Our model considers a pipeline-parallel training setup in which a neural network is partitioned into multiple stages, each stage $i$ assigned to a dedicated GPU. The training data is further divided into a sequence of micro-batches, indexed by $j$. For each micro-batch, every stage must perform three distinct **computational operations** ($c$): a **Forward pass (F)**, a **Backward pass for activations (B)**, and a **Backward pass for weights (W)**.

To manage GPU memory efficiently, we incorporate two additional data-transfer operations: **Offload (O)**, which transfers activation memory from GPU to CPU, and **Reload (R)**, which restores it

when needed. The core objective of our model is to schedule these five types of events, F, B, W, O, and R, in order to minimize the overall makespan.

## 3.2 Decision Variables and Objective

Our formulation is built around a set of decision variables that together define a complete schedule. The timing of the schedule is captured by continuous variables: $E_{(i,j,c)}$ denotes the end time of a computational operation, while $O_{(i,j,c)}$ and $R_{(i,j,c)}$ represent the start times of activation offload and reload operations, respectively. The key strategic decision is modeled by the binary variable $W_{(i,j,c)}$, which indicates whether the activation from operation $(i,j,c)$ is offloaded ($W_{(i,j,c)} = 1$) or retained in GPU memory ($W_{(i,j,c)} = 0$).

To ensure the schedule is physically realizable, we introduce auxiliary binary variables that enforce ordering constraints and resolve resource conflicts. On the GPU's computational core, the variable $P_{(i,j,c)\to(i,j',c')}$ serializes any two computational operations: $P_{(i,j,c)\to(i,j',c')} = 1$ indicates that operation $(i,j,c)$ must be completed before $(i,j',c')$, and $P_{(i,j,c)\to(i,j',c')} = 0$ otherwise. For communication between GPU and CPU, $K_{(i,j,c)\to(i,j',c')}$ and $L_{(i,j,c)\to(i,j',c')}$ sequence pairs of offload and reload operations, respectively, while $H_{(i,j,c)\to(i,j',c')}$ establishes the order between offload and reload events that share the same communication channel.

Dependencies between computation and data transfers are enforced via $M_{(i,j,c)\to(i,j',c')}$ and $N_{(i,j,c)\to(i,j',c')}$, which guarantee that computations begin only after the required data has been produced or reloaded. A value of 1 for any precedence variable indicates that the first event in the subscript must complete before the second begins.

The objective of the MILP is to minimize the makespan of pipeline execution across all stages, represented by the continuous variable $C$. When using post-validation, as suggested in (Qi et al., 2023), the value is determined by the maximum elapsed time from the start of the first operation to the completion of the final operation in each stage. If post-validation is not used, the value is calculated from the start of the first operation in the entire process to the completion of the final operation. (Eq. 3, 4).

## 3.3 Key Constraints

The model is governed by a set of constraints that guarantee the resulting schedule is both valid and physically realizable. We summarize these constraints below, while the complete mathematical formulations are deferred to Appendix C.1 due to space limitations.

- **Data-Dependency Constraints:** These constraints enforce the fundamental dataflow of pipeline parallelism. The forward pass of micro-batch $j$ on stage $i$ can only begin after the forward pass on stage $i-1$ is completed (Eq. 5). Similarly, the backward pass of stage $i$ depends on the completion of the backward pass on stage $i+1$ (Eq. 6). Within a single stage, each micro-batch must follow the strict sequence Forward $\to$ Backward-activation $\to$ Backward-weight (Eq. 8).

- **Resource Exclusivity Constraints:** A GPU can execute only one computational operation at a time, and the communication channel between GPU and CPU can handle only one offload or reload at a time. We enforce exclusivity using the standard Big-M method (Trespalacios & Grossmann, 2015) with binary precedence variables (e.g., $P_{(i,j,c)\to(i,j',c')}$), ensuring that no two operations assigned to the same resource overlap in time (Eq. 7, 10–13).

- **Memory Capacity Constraints:** To respect the GPU's physical capacity $M_i^{\text{limit}}$, we track dynamic memory usage at each stage. Memory consumption evolves as computations complete (contributing $\Delta_{(i,j,c)}$) and as activations are offloaded or reloaded (contributing $\Gamma_{(i,j,c)}$). Constraint (Eq. 9) directly couples the operational schedule with memory feasibility, ensuring that no GPU exceeds its memory limit at any point in time.

- **Synchronization Constraints:** These constraints coordinate the interaction between computation and data transfers. A reload operation $R_{(i,j,c)}$ must complete before the computation that consumes the corresponding activation can begin. Conversely, an offload operation $O_{(i,j,c)}$ can only start after the associated forward pass has finished and produced the activation data (Eq. 14–17).

- **Topology-Aware Offload Constraints:** As observed in (Wan et al., 2025), offloading efficiency depends on the interconnect topology between GPUs and CPUs. For example, on A100 systems,

two GPUs may share a PCIe switch, preventing simultaneous independent offloads. By contrast, H100 GPUs are directly linked to the CPU via independent PCIe connections, enabling concurrent offloads without interference. We use constraints (Eq 18) to control this in the MILP model.

By solving this MILP, we obtain a globally optimal schedule that balances computation, communication, and memory pressure to minimize training time. To reduce complexity for practical solvers, we introduce simplifications such as fixing the processing order of symmetric micro-batches, which significantly prunes the search space without affecting optimality.

# 4 OptPipe: An Efficient Implementation of MILP-based Scheduling

In this section, we present OptPipe, an efficient implementatoin MILP-based scheduling approach for real-world training systems.

## 4.1 Practical Optimizations for MILP Solving

To make our MILP formulation practical for large-scale pipeline parallelism, we incorporate a set of solver-level optimizations. These include variable fixing, cut generation, redundancy elimination, cached schedule strategy, and warm-starting with initial solutions. Together, these strategies significantly reduce solving overhead without compromising solution quality. Throughout this section, we illustrate the techniques using the precedence variables $P_{(i,j,c)}$ as examples, though the same principles apply equally to the other types of ordering variables.

### 4.1.1 Redundancy Elimination

**Fixed Micro-batch Order and Symmetry Breaking** Since micro-batches are symmetric, we can fix their processing order to eliminate redundant scheduling possibilities. For pairs of precedence variables like $P$, we only define variables for one direction and derive the other logically. Then, we have following equations:

$$P_{(i,j,c) \to (i,j',c)} = 1, \quad \forall i, j' > j, c$$
$$P_{(i,j',c') \to (i,j,c)} = 1 - P_{(i,j,c) \to (i,j',c')}, \forall i \tag{1}$$

**Remove Indirectly Determined Binary Variables** To reduce model size and improve solver efficiency, we exploit structural properties of the binary variables that encode ordering relationships. These variables exhibit both symmetry and transitivity, which can be leveraged to eliminate redundancy.

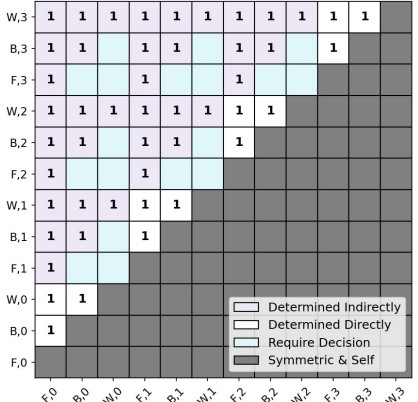

As illustrated in Figure 3, we retain only the upper-triangular portion of the precedence matrix. By symmetry, each ordering variable has a complementary counterpart: if $P_{(i,j,c) \to (i,j',c')}$ indicates that operation $(i, j, c)$ precedes $(i, j', c')$, then $P_{(i,j',c') \to (i,j,c)}$ is its negation. Thus, variables in the lower-triangular region (shown in gray) are unnecessary and can be inferred directly.

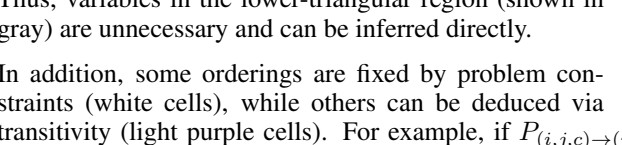

Figure 3: Determined and Undetermined Variables

In addition, some orderings are fixed by problem constraints (white cells), while others can be deduced via transitivity (light purple cells). For example, if $P_{(i,j,c) \to (i,j',c')} = 1$ and $P_{(i,j',c') \to (i,j'',c'')} = 1$, then $P_{(i,j,c) \to (i,j'',c'')} = 1$ must also hold. Such indirectly determined variables need not be introduced into the model explicitly. Consequently, only the blue cells in Figure 3 correspond to precedence variables that require explicit solver decisions.

### 4.1.2 Triangle Inequality Cuts

We introduce a cutting-plane strategy to further accelerate MILP solving. The goal is to narrow the gap between the MILP and its Linear Programming (LP) relaxation by leveraging the transitive

property of sequencing variables. This yields a family of valid inequalities, commonly referred to as triangle inequality cuts, that are widely used in scheduling problems (Fleming et al., 2013; Ascheuer et al., 1993; Oliveira & Pessoa, 2020). We apply these cuts systematically across all precedence variables.

Formally, let the binary variable $P_{(i,j,c)\to(i_1,j_1,c_1)}$ equal 1 if task $(i,j,c)$ precedes task $(i_1,j_1,c_1)$, and 0 otherwise. By transitivity, if task A precedes B and B precedes C, then A must precede C. This logical relationship can be encoded as the linear inequality

$$P_{(i,j,c)\to(i_2,j_2,c_2)} \geq P_{(i,j,c)\to(i_1,j_1,c_1)} + P_{(i_1,j_1,c_1)\to(i_2,j_2,c_2)} - 1,$$

for any three distinct tasks $(i,j,c)$, $(i_1,j_1,c_1)$, and $(i_2,j_2,c_2)$. This constraint enforces consistency in the solution space: if $(i,j,c)$ precedes $(i_1,j_1,c_1)$ and $(i_1,j_1,c_1)$ precedes $(i_2,j_2,c_2)$, then $(i,j,c)$ must precede $(i_2,j_2,c_2)$. By systematically generating such cuts for all relevant task triplets, we tighten the LP relaxation without excluding any integer-feasible schedules, thereby improving the efficiency of the branch-and-cut algorithm.

### 4.1.3 INITIAL SOLUTION STRATEGIES

Finding a good initial solution is crucial for improving the efficiency of MILP solving. Even though the problem of finding a feasible solution itself is NP-hard, in our specific context, a trivial feasible schedule can be obtained by running pure pipeline parallelism with a single micro-batch, which completely eliminates memory pressure. However, this naive approach results in an excessively large makespan and significant idle time ("bubbles") due to the lack of overlap between computations. More advanced schemes such as 1F1B (Fan et al., 2021) and Zero Bubble (Qi et al., 2023) often become infeasible under strict memory budgets, since they do not explicitly account for memory constraints.

PipeOffload (Wan et al., 2025) provides a more suitable baseline for memory-limited scenarios by offloading all forward (F) chunks and combining backward-activation (B) and backward-weight (W) chunks. This strategy guarantees the minimum possible memory usage, but it does not exploit the actual memory limit of the device. In particular, it only schedules a small number of forward chunks before starting the first backward chunk, which leads to suboptimal utilization. Our observations suggest that scheduling more forward chunks in the fill phase can produce higher-quality solutions.

To this end, we propose AdaOffload, an initialization strategy that generates schedules with a denser fill phase. AdaOffload determines the maximize number of forward chunks to place before the first backward chunk at each stage, subject to memory constraints, while following PipeOffload's strategy for the remaining schedule. A detailed description of AdaOffload is provided in Appendix D. While the makespan produced by AdaOffload can outperform PipeOffload, it significantly enhances the solving efficiency of the MILP. In Figure 4, we present a toy example comparing different offloaded pipeline parallelism strategies, including the optimal strategy. The example assumes that all processing times are equal, and that the memory can hold up to three activations at once. This simplified setup enables a direct comparison of the strategies under ideal conditions, as illustrated in the figure: AdaOffload achieves a lower makespan while maintaining a similar module during the fill phase, thereby providing a better warm start for solving MILP problems.

### 4.2 CACHED SCHEDULE STRATEGY

Since solving the MILP can be time-consuming, it is desirable to reuse previously solved schedules that can be quickly adapted to new settings. A major challenge, however, is that the estimated parameters in the MILP, such as computation time, communication latency, and memory usage, can vary stochastically across runs or hardware environments.

To address this, we introduce a crude schedule strategy based on discretization. Specifically, we discretize the estimated parameters for computation, communication, and memory into proportional values. When solving a new instance, we search for the most similar crude schedule in this discretized space and use it to initialize the solver. This warm-start procedure improves the quality of initial strategies while ensuring reusability across different problem instances. If no cached schedule is sufficiently similar, we fall back to the default initialization strategy described earlier.

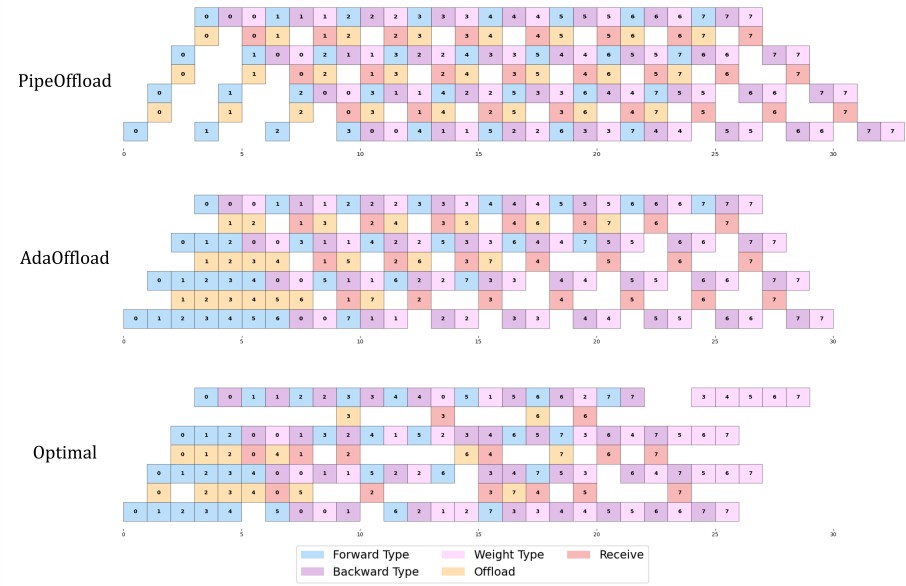

Figure 4: A toy example for illustration. Each block in the figure represents a distinct operation type, with the colors indicating different stages. The x-axis represents time, and the y-axis shows different stages in the pipeline parallelism.

### 4.3 ONLINE SCHEDULING

The time required to find an optimal schedule grows rapidly with the number of training stages, due to the NP-hard nature of the scheduling problem. To mitigate this drawback, we propose solving the scheduling problem dynamically during the training phase of the LLM. This is feasible because the solver runs on CPUs, while training primarily utilizes GPUs.

By leveraging the solver's callback functionality, which can detect improved solutions and customize the workflow, we can continuously update the schedule during training. As the solver discovers better schedules, the system can adopt them without interrupting training. An additional advantage of this framework is adaptability: when estimated parameters (e.g., computation or communication times) change significantly, the scheduler can adjust accordingly, preventing the application of outdated or suboptimal schedules.

## 5 EXPERIMENTS

### 5.1 EXPERIMENT SETTINGS

**Implementation Details** We implemented our method on top of the open-source Megatron-LM framework (Narayanan et al., 2021), incorporating the implementations of Zero Bubble (ZB) pipeline parallelism and PipeOffload (Qi et al., 2023; Wan et al., 2025). Following ZB, we perform a few warm-up iterations to estimate key pipeline parameters, including $T_F$, $T_B$, $T_{comm}$, and $T_{offload}$. For these estimation runs, we adopt PipeOffload due to its minimal memory footprint. The resulting Mixed-Integer Linear Programming (MILP) problem is then solved using Gurobi (Gurobi Optimization, 2020) as the backend solver.

**Infrastructure and Configuration** Our experiments are conducted on a cluster with up to 16 NVIDIA H100 GPUs. We evaluate models with architectures analogous to GPT-3, ensuring a representative setting for large-scale LLM training. Detailed model configurations are provided in Appendix B.

**Baseline** We compare our approach against 5 pipeline parallelism baselines: 1F1B (Fan et al., 2021), 1F1B-Interleaved (1F1B-I) (Narayanan et al., 2021), Zero Bubble (ZB) (Qi et al., 2023), Zero Bubble-V (ZB-V) (Qi et al., 2024), and PipeOffload (Wan et al., 2025).

## 5.2 EVALUATION

To evaluate the quality of scheduling strategies, we ran 120 iterations for each configuration and used the average elapsed time of the last 100 iterations as the primary performance metric. The complete results are presented in Table 1, covering a comprehensive range of GPU counts, model parameter sizes, micro-batch numbers, and micro-batch sizes. In each experiment, we use AdaOffload to provide a initial solution to Gurobi, which can significant improve solving efficiency.

As shown in Table 1, in memory-rich scenarios, OptPipe achieves performance comparable to 1F1B, 1F1B-I, ZB, and ZB-V, while being more than 30% faster than PipeOffload. In contrast, under memory-limited settings, such as the 1.5B model with a micro-batch size of 32, where all baselines except PipeOffload encounter out-of-memory (OOM) errors, OptPipe still outperforms PipeOffload by more than 20%. These results demonstrate that OptPipe delivers both superior performance and robustness across a wide range of scenarios.

| Params | Number | Size | 1F1B | 1F1B-I | ZB | ZB-V | PipeOffload | OptPipe (ours) |
|--------|--------|------|------|--------|----|----|-----|-----|
| | | | **GPU NUMBER: 4** | | | | | |
| 1.5B | 8 | 4 | 1423.24 | 2193.50 | **1254.61** | 1363.93 | 1651.83 | 1283.00 |
| | | 8 | 1475.22 | 2218.30 | 1312.17 | 1375.72 | 2025.03 | **1302.40** |
| | | 16 | 1579.51 | OOM | OOM | **1517.89** | 4184.03 | 1674.37 |
| | | 24 | OOM | OOM | OOM | OOM | 5402.70 | **2517.53** |
| | | 32 | OOM | OOM | OOM | OOM | 7176.87 | **4361.82** |
| | 16 | 4 | 1949.70 | 2034.90 | **1851.26** | 2098.33 | 4028.40 | 2389.93 |
| | | 8 | **1883.20** | 1996.70 | 1962.40 | 2112.65 | 4006.87 | 2350.43 |
| | | 16 | **2689.30** | OOM | OOM | OOM | 6911.27 | 2951.70 |
| | | 32 | OOM | OOM | OOM | OOM | 10321.45 | **7135.41** |
| 3.6B | 8 | 4 | 1157.43 | 1146.40 | **1106.40** | 1189.70 | 1938.73 | 1358.53 |
| | | 8 | 1540.00 | 2791.40 | 1507.40 | **1368.70** | 2994.23 | 1588.50 |
| | | 16 | OOM | OOM | OOM | OOM | 5123.34 | **2144.76** |
| | 16 | 4 | 1632.40 | 1612.45 | **1493.45** | 1579.43 | 2609.03 | 1828.23 |
| | | 8 | **1977.22** | 2011.13 | OOM | 1994.32 | 4029.46 | 2137.71 |
| | | 16 | OOM | OOM | OOM | OOM | 6894.68 | **2886.29** |
| | | | **GPU NUMBER: 8** | | | | | |
| 7.1B | 16 | 1 | 1983.40 | 2192.00 | **1927.50** | 2826.40 | 3033.87 | 1929.87 |
| | | 2 | 2147.00 | 2526.10 | **2029.50** | 2974.30 | 3741.43 | 2369.13 |
| | | 4 | **2198.30** | 2608.60 | 2305.80 | 2997.30 | 5345.13 | 2767.60 |
| | | 8 | OOM | OOM | OOM | OOM | 7131.31 | **3913.10** |
| | | 16 | OOM | OOM | OOM | OOM | 15152.12 | **11747.92** |
| | 32 | 1 | 3709.30 | 4007.45 | **3662.90** | 4808.90 | 5796.67 | 4470.33 |
| | | 2 | 3836.60 | 4235.90 | **3730.20** | 4909.10 | 5878.30 | 4639.87 |
| | | 4 | 3843.10 | 4543.70 | **3723.70** | 5053.50 | 10012.23 | 4666.33 |
| | | 8 | OOM | OOM | OOM | OOM | 20981.80 | **15793.80** |
| | | 16 | OOM | OOM | OOM | OOM | 41254.60 | **31445.20** |
| | | | **GPU NUMBER: 16** | | | | | |
| 14.2B | 32 | 1 | 2744.34 | 3054.29 | **2591.14** | 3676.36 | 3971.16 | 2602.56 |
| | | 2 | 2857.77 | 3506.94 | **2785.44** | 4108.14 | 4983.86 | 2812.45 |
| | | 4 | **3047.62** | 3412.21 | 3058.63 | 4115.23 | 7446.86 | 3124.42 |
| | | 8 | OOM | OOM | OOM | OOM | 34134.23 | **20140.23** |
| | | 16 | OOM | OOM | OOM | OOM | 42156.53 | **35175.32** |
| | 64 | 1 | 5020.38 | 5352.19 | **4845.31** | 6322.89 | 8047.44 | 5031.53 |
| | | 2 | 5160.82 | 5885.96 | 5187.24 | 6684.25 | 8145.83 | **5123.43** |
| | | 4 | **5034.34** | 6131.95 | 5130.98 | 6972.76 | 13281.90 | 5082.24 |
| | | 8 | OOM | OOM | OOM | OOM | 41589.35 | **31593.20** |
| | | 16 | OOM | OOM | OOM | OOM | 50124.41 | **41679.20** |

Table 1: Performance Comparison of Pipeline Parallelism Methods. This table reports the average iteration time (ms). For each row, the **best** result is shown in bold, and the second-best is underlined. The columns indicate: **Params** (model parameter size), **Number** (micro-batch number), **Size** (micro-batch size), and the performance of different pipeline parallelism methods. Rows labeled GPU Number K correspond to experiments using $K$ GPUs. "OOM" means Out of Memory.

**Solving Time** We set the time limits of solving MILP for each scenario (300s for 4/8 GPUs and 1000s for 16 GPUs) and select the best feasible solution within the limit. For small cases (e.g., 4 GPUs), the commercial solver typically finds the optimal solution, while for larger cases (e.g., 16 GPUs), it often only reaches the time limit. Thanks to our heuristics, however, we can always obtain high-quality solutions within the limit, even if they are not optimal. Moreover, these MILP problems can be solved offline and cached, or updated online during training, ensuring that runtime does not become a bottleneck in practice.

## 5.3 IN-DEPTH ANALYSIS

This section analyzes the performance differences between OptPipe and PipeOffload, as both methods are specifically designed to operate under memory constraints.

**Memory Usage Analysis** Figure 5 illustrates the key mechanism underlying OptPipe's superior performance: a more effective trade-off between memory usage and efficiency. It consistently maintains higher average (AVG) and maximum (MAX) memory utilization, leveraging available capacity to improve throughput under strict limits. In contrast, PipeOffload is more conservative, leaving memory underutilized and thereby reducing efficiency. OptPipe effectively converts idle memory into performance gains, demonstrating more efficient time–memory trade-off management.

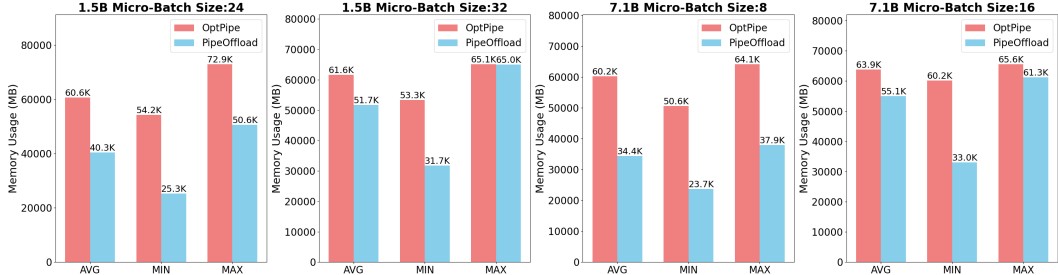

Figure 5: Memory Usage Comparison between PipeOffload and OptPipe. Average, minimum, and maximum device memory usage across different model sizes and micro-batch sizes.

**Analysis of Performance with Varying Micro-Batch Numbers** Figure 6 compares elapsed time under varying micro-batch counts (16–256). OptPipe consistently outperforms PipeOffload across all settings, with efficiency gains growing as workload increases. In the most demanding case (micro-batch size 8, 256 micro-batches), OptPipe reduces training time by about 17%, demonstrating its effectiveness in large-scale scenarios where minimizing device idle time is critical.

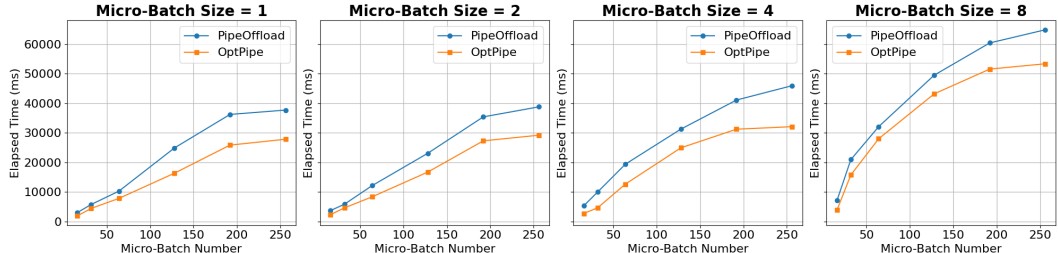

Figure 6: Elapsed time comparison between PipeOffload and OptPipe under varying micro-batch numbers for an 8-GPU setup and a 7.1B LLM model.

## 6 CONCLUSION

In this work, we presented OptPipe, a framework for optimizing pipeline parallelism with activation offloading. We modeled scheduling as a MILP problem and introduced practical techniques to make the approach feasible in large-scale training. OptPipe significantly improves the efficiency of LLM training, underscoring the importance of refined scheduling for pipeline parallelism. Future work will explore further accelerating the solver and enhancing model robustness, as well as extending OptPipe to scenarios that combine pipeline parallelism with other parallelization schemes through communication–computation overlap.

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

## A  LLM USAGE DECLARATION

In this submission, we used a Large Language Model (LLM) solely for language refinement and text polishing. The LLM was employed to enhance the clarity, flow, and readability of the manuscript, but it did not contribute to the ideation, analysis, or generation of scientific content. All ideas, interpretations, and results presented in the paper are solely the work of the authors. No content generated by the LLM was used to fabricate facts or contribute to the research findings.

## B  CONFIGURATION

**Model Configuration**  We present the model configuration for different model sizes. The configurations are summarized in the Table B. These configurations represent the key hyperparameter that define the architecture and performance of the models with varying sizes: 1.5B, 3.6B, 7.1B, and 14.2B parameters. Each size corresponds to a different number of layers, hidden units, and other model-specific settings. For each setting, we set sequence length as 1024 and post validation is used to further improve efficiency.

| Parameter | Model Sizes | | | |
|---|---|---|---|---|
| | **1.5B** | **3.6B** | **7.1B** | **14.2B** |
| `num-layers` | 128 | 128 | 256 | 256 |
| `hidden-size` | 2048 | 2048 | 128 | 128 |
| `ffn-hidden-size` | 4096 | 4096 | 4096 | 4096 |
| `num-attention-heads` | 16 | 16 | 16 | 16 |
| `num-query-groups` | 8 | 8 | 8 | 8 |

Table 2: Model Configuration for Different Sizes

## C  MATHEMATICAL MODELS

### C.1  MIXED INTEGER LINEAR PROGRAMMING FORMULATION

**Notation**  We define the following indices, decision variables, and parameters for the model.

- **Indices**:
  - $i$: Index for the $i$-th stage (i.e., the $i$-th GPU).
  - $j$: Index for the $j$-th micro-batch. We assume each micro-batch has identical parameters.
  - $c$: Index for the type of operation, where $c \in \{F, B, W\}$ represents Forward, Backward for activation, and Backward for weights, respectively.
- **Terminology**:
  - **Offload**: Transferring activation memory from a GPU to the CPU.
  - **Reload**: Transferring activation memory from the CPU back to a GPU.
  - **Operation**: A computation step, which can be Forward (F), Backward for activation (B), or Backward for weights (W).

**Decision Variables**

$P_{(i,j,c)\to(i,j',c')}$: A binary variable that is 1 if the operation $(i,j,c)$ is processed before operation $(i,j',c')$, and 0 otherwise.

$K_{(i,j,c)\to(i,j',c')}$: A binary variable that is 1 if the offloading for $(i,j,c)$ starts before the offloading for $(i,j',c')$, and 0 otherwise.

$L_{(i,j,c)\to(i,j',c')}$: A binary variable that is 1 if the reloading for $(i,j,c)$ starts before the reloading for $(i,j',c')$, and 0 otherwise.

$M_{(i,j,c)\to(i,j',c')}$: A binary variable that is 1 if the offloading for $(i,j,c)$ starts before the processing of $(i,j',c')$, and 0 otherwise.

$N_{(i,j,c)\to(i,j',c')}$: A binary variable that is 1 if the reloading for $(i,j,c)$ starts before the processing of $(i,j',c')$, and 0 otherwise.

$H_{(i,j,c)\to(i,j',c')}$: A binary variable that is 1 if the offloading for $(i,j,c)$ starts before the reloading for $(i,j',c')$, and 0 otherwise.

$E_{(i,j,c)}$: A continuous variable representing the end time of the operation $(i,j,c)$.

$O_{(i,j,c)}$: A continuous variable representing the start time of the activation offload for $(i,j,c)$.

$R_{(i,j,c)}$: A continuous variable representing the start time of the activation reload for $(i,j,c)$.

$W_{(i,j,c)}$: A binary variable that is 1 if the activation for $(i,j,c)$ is offloaded, and 0 otherwise.

$C$: A continuous variable representing the total time cost (makespan).

**Parameters**

$\Delta_{(i,j,c)}$: The amount of memory change after completing operation $(i,j,c)$. This satisfies: $\Delta_{(i,j,F)} + \Delta_{(i,j,B)} + \Delta_{(i,j,W)} = 0$, with $\Delta_{(i,j,F)} > 0$, $\Delta_{(i,j,B)} < 0$, and $\Delta_{(i,j,W)} < 0$.

$\Gamma_{(i,j,c)}$: The amount of memory occupied by the activations of $(i,j,c)$.

$T_{(i,j,c)}$: The processing time for operation $(i,j,c)$.

$T_{\text{comm}}$: The communication time for transferring activations between adjacent GPUs.

$T_{\text{offload}}$: The time required to offload or reload the activations of a single operation between a GPU and the CPU.

$M_i^{\text{limit}}$: The memory capacity of the $i$-th GPU.

**Model Formulation**    The objective is to minimize the total pipeline execution time, $C$.

$$\min \quad C \tag{2}$$

This is subject to the following constraints:

- **Makespan Definition**: When using Post Validation, he total cost $C$ is the maximum time span from the start of the first operation to the end of the last operation on any stage in Pipeline Parallelism.

$$C \geq E_{(i,m,W)} - (E_{(i,1,F)} - T_{(i,1,F)}), \qquad \forall i \tag{3}$$

where $m$ is the last micro-batch.Instead, we define $C$ is the maximum time span from the first operation to the end operation over whole schedule.

$$C \geq E_{(i,j,W)} - (E_{(1,1,F)} - T_{(1,1,F)}), \qquad \forall i,j \tag{4}$$

- **Pipeline Data Dependencies**: The Forward pass on GPU $i$ must wait for the Forward pass on GPU $i-1$. The Backward pass on GPU $i$ must wait for the Backward pass on GPU $i+1$.

$$E_{(i,j,F)} \geq E_{(i-1,j,F)} + T_{\text{comm}} + T_{(i,j,F)}, \qquad \forall i,j \tag{5}$$

$$E_{(i,j,B)} \geq E_{(i+1,j,B)} + T_{\text{comm}} + T_{(i,j,B)}, \qquad \forall i,j \tag{6}$$

- **Intra-GPU Operation Sequencing**: Only one operation can be active on a single GPU at any time. A large constant $\mathcal{M}$ is used for the Big-M method.

$$E_{(i,j,c)} \geq E_{(i,j',c')} + T_{(i,j,c)} - P_{(i,j,c)\to(i,j',c')} \cdot \mathcal{M}, \qquad \forall i,j,j' \neq j, c,c' \neq c \tag{7}$$

- **F-B-W Order**: For a given micro-batch on a given GPU, the Forward, Backward-activation, and Backward-weight operations must be executed in order.

$$P_{(i,j,F)\to(i,j,B)} = 1, P_{(i,j,B)\to(i,j,W)} = 1, \qquad \forall i,j \tag{8}$$

- **Memory Capacity Constraint**: The total memory used during any operation $(i,j',c')$ must not exceed the GPU's memory limit.

$$M_i^{\text{limit}} \geq \Delta_{(i,j',c')} + \sum_{j,c} \Delta_{(i,j,c)} P_{(i,j,c)\to(i,j',c')}$$
$$- \Gamma_{(i,j,c)} M_{(i,j,c)\to(i,j',c')} + \Gamma_{(i,j,c)} N_{(i,j,c)\to(i,j',c')}, \qquad \forall i,j',c' \tag{9}$$

- **Offload/Reload Sequencing**: If an operation $(i, j', c')$ is chosen for offloading, its offload and reload phases must be sequenced with respect to other offloads and reloads on the same GPU.

$$O_{(i,j,c)} \geq O_{(i,j',c')} + T_{\text{offload}} - K_{(i,j,c) \to (i,j',c')} \cdot \mathcal{M} - (1 - W_{(i,j',c')}) \cdot \mathcal{M} \tag{10}$$

$$R_{(i,j,c)} \geq R_{(i,j',c')} + T_{\text{offload}} - L_{(i,j,c) \to (i,j',c')} \cdot \mathcal{M} - (1 - W_{(i,j',c')}) \cdot \mathcal{M} \tag{11}$$

$$R_{(i,j,c)} \geq O_{(i,j',c')} + T_{\text{offload}} - (1 - H_{(i,j',c') \to (i,j,c)}) \cdot \mathcal{M} - (1 - W_{(i,j',c')}) \cdot \mathcal{M} \tag{12}$$

$$O_{(i,j,c)} \geq R_{(i,j',c')} + T_{\text{offload}} - H_{(i,j,c) \to (i,j',c')} \cdot \mathcal{M} - (1 - W_{(i,j',c')}) \cdot \mathcal{M} \tag{13}$$

for all $i, j, j' \neq j, c, c' \neq c$.

- **Offload/Reload and Operation Synchronization**: If offloading is performed, the timing must respect the dependencies with computational operations.

$$E_{(i,j,c)} - T_{\text{comm}} \geq O_{(i,j',c')} + T_{\text{offload}} - (1 - M_{(i,j',c') \to (i,j,c)}) \cdot \mathcal{M} - (1 - W_{(i,j',c')}) \cdot \mathcal{M} \tag{14}$$

$$E_{(i,j,c)} \geq R_{(i,j',c')} + T_{(i,j,c)} - (1 - N_{(i,j',c') \to (i,j,c)}) \cdot \mathcal{M} - (1 - W_{(i,j',c')}) \cdot \mathcal{M} \tag{15}$$

$$R_{(i,j',c')} \geq E_{(i,j,c)} - N_{(i,j',c') \to (i,j,c)} \cdot \mathcal{M} - (1 - W_{(i,j',c')}) \cdot \mathcal{M} \tag{16}$$

for all $i, j, j' \neq j, c, c' \neq c$.

- **Offload Choice Consistency**: An operation cannot have a precedence relationship with an offload/reload if it is not chosen to be offloaded.

$$M_{(i,j,c) \to (i,j',c')} \leq W_{(i,j,c)}, N_{(i,j,c) \to (i,j',c')} \leq W_{(i,j,c)}, \qquad \forall i, j, c, j', c' \tag{17}$$

- **Topology-Aware Constraints**: If multiple GPUs are connected to the CPU through the same PCIe switch, we require that the offload and reload processes do not occur simultaneously on these GPUs in order to manage the offloading time effectively.

$$O_{(i_2,j,c)} \geq O_{(i_1,j,c)} + T_{\text{offload}} - L_{(i_2,j,c) \to (i_1,j,c)} \tag{18}$$

$$R_{(i_2,j,c)} \geq R_{(i_1,j,c)} + T_{\text{offload}} - L_{(i_2,j,c) \to (i_1,j,c)} \tag{19}$$

, where $i_1, i_2$ are devices connected to the CPU through same PCIe switch.

- **Fixed Micro-batch Order**: Since micro-batches are symmetric, we can fix their processing order to eliminate redundant scheduling possibilities.

$$P_{(i,j,c) \to (i,j',c)} = 1, \quad \forall j' > j$$

- **Symmetry Breaking for Binary Variables**: For pairs of precedence variables like $P$ and $H$, we only define variables for one direction and derive the other logically.

$$P_{(i,j',c') \to (i,j,c)} = 1 - P_{(i,j,c) \to (i,j',c')}$$

$$H_{(i,j',c') \to (i,j,c)} = 1 - H_{(i,j,c) \to (i,j',c')}$$

# D  ADAOFFLOAD

This method is inspired by **PipeOffload**, but with a key difference: it considers the availability of more memory compared to what PipeOffload typically requires. In cases where memory is less constrained, the approach strives to fill as many forward tasks (F) as possible before the first backward task (B) begins. The objective is to make the fill phase more **dense**, bringing it closer to an optimal distribution. This not only improves the pipeline's efficiency but also aids in solving MILP problems by making the scheduling more favorable for the solver.

When memory is limited, or the offloading time becomes excessively long, the method effectively **falls back to the PipeOffload rules**. In such cases, it behaves similarly to the original PipeOffload approach, where the focus is on reducing offload times and maintaining a balanced schedule.

---

**Algorithm 1** AdaOffload: An Denser Initial Pipeline Schedule

---

**Require:**
1: $G = (V, E)$ {Computational graph (F/B/W tasks, Offload (O) task and Reload (R) task.}
2: $\text{cost}_{\text{comm}}$ {Inter-stage communication latency}
3: $n_{\text{stages}}, n_{\text{mb}}$ {Number of stages and micro-batches}
4: $T$ {Tolerance of delaying the first B task in each stage}

5: /* Step 1: Compute the earliest start time for each backward task */
6: **for** $s = 0$ **to** $n_{\text{stages}} - 1$ **do**
7:     $T_F^{\max} \leftarrow \max$ runtime of all forward tasks in stage $s$
8:     $T_B^{\max} \leftarrow \max$ runtime of all backward tasks in stage $s$
9:     $T_W^{\max} \leftarrow \max$ runtime of all weight-update tasks in stage $s$
10:     $T_{\text{offload}}^{\max} \leftarrow \max$ runtime of all activation offloading tasks in stage $s$
11:     Compute $\text{EstStart}(B_{s,0})$ as the earliest start time for backward tasks

12: /* Step 2: Schedule forward tasks and backward tasks for overlap */
13: **for** $s = 0$ **to** $n_{\text{stages}} - 1$ **do**
14:     /* Fill forward tasks (F) as much as possible before backward tasks */
15:     **for** $m = 0$ **to** $n_{\text{mb}} - 1$ **do**
16:         **if** $\max(\text{complete\_time}(F_{s,m-1}), \text{complete\_time}(O_{s,m-1}) + T_{\text{offload}}^{\max}) \geq \text{EstStart}(B_{s,0}) + T$
    **then**
17:         **BREAK**{Schedule forward tasks as early as possible}
18:         $\text{start}(F_{s,m}) \leftarrow \max(\text{complete\_time}(F_{s,m-1}), \text{complete\_time}(O_{s,m-1}), \text{complete\_time}(F_{s-1,n_{\text{mb}}}) + \text{cost}_{\text{comm}})$
19:         $\text{complete\_time}(O_{s,m}) \leftarrow \text{complete\_time}(F_{s,m}) + T_{\text{offload}}^{\max}$
20:         $\text{complete\_time}(F_{s,m}) \leftarrow \text{start}(F_{s,m}) + \text{runtime}(F_{s,m})$
21:         /* Slightly delay backward tasks to fit more forward tasks */
22:         $\text{start}(B_{s,0}) \leftarrow \max(\text{complete\_time}(F_{s,m}), \text{EstStart}(B_{s,0}), \text{complete\_time}(O_{s,m}) + T_{\text{offload}}^{\max})$
23:         $\text{complete\_time}(B_{s,0}) \leftarrow \text{start}(B_{s,0}) + \text{runtime}(B_{s,0})$

24: /* Step 3: Finish scheduling using PipeOffload-style rules */
25: **for** $s = 0$ **to** $n_{\text{stages}} - 1$ **do**
26:     Schedule the remaining backward and weight-update tasks based on PipeOffload strategy
27:     Allow overlap between B and W tasks for better pipeline utilization

28: /* Compute makespan and task order */
29: $M \leftarrow \max_{v \in V}\{\text{complete\_time}(v)\}$
30: Topological order $\prec$ from non-decreasing $\text{start}(v)$
31: **Return** $M, \mathcal{S} = \{\text{start}(v), \prec\}$

---

