# OpenReview forum: "OptPipe: Memory- and Scheduling-Optimized Pipeline Parallelism for LLM Training"
_ICLR.cc/2026/Conference — ICLR 2026 Conference Withdrawn Submission_

### Official Review · Reviewer_PRfU · 2025-10-20

**Soundness:** 2
**Presentation:** 2
**Contribution:** 2
**Rating:** 2
**Confidence:** 3

**Summary:**

This paper proposes a method that solves the pipeline scheduling problem using mixed-integer linear programming (MILP), treating activation offloading as a decision variable. It models whether activations are offloaded or retained in GPU memory and enforces constraints on data dependencies, resource exclusivity, memory capacity, synchronization, and GPU–CPU topology. Because solving the MILP can be computationally expensive, the paper introduces several accelerations: variable fixing, cut generation, redundancy elimination, a cached scheduling strategy, and warm starts with initial solutions. The method is evaluated on up to 16 NVIDIA H100 GPUs with GPT-3–like architectures, demonstrating schedules that achieve speedups and avoid out-of-memory errors compared to baselines. The baselines include five pipeline-parallelism methods: 1F1B, 1F1B-Interleaved, ZeroBubble, ZeroBubble-V, and PipeOffload. Results show >30% faster performance than PipeOffload in memory-rich settings and >20% faster in memory-limited settings. Since the MILP solver can sometimes run for a long time, a time limit is imposed and the best solution found within that limit is used. The proposed method consumes more memory than PipeOffload to realize its speedups, illustrating a clear time–memory trade-off.

**Strengths:**

- This paper targets the important problem of finding an optimal schedule for pipeline-parallel training of large language models.

- It compares its effectiveness against recent scheduling methods such as ZeroBubble, ZeroBubble-V, and PipeOffload.

- It formulates optimal pipeline scheduling with activation offloading as a mixed-integer linear program (MILP), which enables optimality checks and strengthens the theoretical grounding.

- The illustrative figures explaining the method and background are very helpful for understanding the content.

- The paper includes memory-usage analysis and a sensitivity study over micro-batch size, clarifying the time–memory trade-off and showing results beyond a single “sweet spot.”

**Weaknesses:**

Although the paper formulates pipeline scheduling with activation offloading and solves it via mixed-integer linear programming, I have several concerns about the transparency and completeness of the experiments.

- The hardware configuration is not stated in sufficient detail—for example, the interconnect bandwidth between nodes/servers used in the experiments.

- Some optimization techniques (e.g., topology-aware offload constraints) appear very similar to PipeOffload’s idea of selecting based on the ratio between activation transfer round-trip time and compute time.

- The paper does not compare against baselines that combine multiple parallelism strategies (e.g., intra-node tensor parallelism + inter-node pipeline parallelism). To demonstrate effectiveness, the method should outperform widely used configurations, especially since tensor parallelism is common within a node.

- The resulting schedules produced by the method are not clearly presented, and the insights we can draw from them are missing. Visualizing and/or analyzing the found schedules would help explain where the speedups come from.

- Analytical formulas for activation memory and pipeline bubbles under the proposed method are not provided. Deriving these and comparing them with baseline formulas would clarify the method’s advantages analytically.

- The evaluation covers only a single model family (GPT-3–like). It lacks results for architectures that have recently gained attention, such as mixture-of-experts (MoE), and for strong contemporary models like Qwen and DeepSeek.

**Questions:**

- Can you provide full hardware specifications, including interconnect bandwidth between nodes/servers used in the experiments?

- How do your topology-aware offload constraints differ from PipeOffload’s ratio-based approach (activation transfer round-trip time vs. compute time)?

- Can you compare against configurations that combine intra-node tensor parallelism with inter-node pipeline parallelism? Do you outperform these widely used setups?

- Can you present the resulting schedules (e.g., visualizations) and analyze them to explain where the speedups originate?

- Can you derive and report formulas for activation memory and pipeline bubbles under your method, and compare them to baseline formulas?

- Can you include results beyond GPT-3–like models—e.g., MoE architectures—and stronger contemporaries such as Qwen and DeepSeek?

**Details Of Ethics Concerns:**

There are no special ethical concerns for this paper.

---

### Official Review · Reviewer_8B3H · 2025-10-27

**Soundness:** 2
**Presentation:** 3
**Contribution:** 3
**Rating:** 6
**Confidence:** 4

**Summary:**

This work introduces a pipeline-parallel training scheduler that jointly optimizes computation order, memory usage, and GPU–CPU activation transfers. The authors formulate scheduling as a Mixed-Integer Linear Program (MILP) involving: binary variables for offloading decisions and precedence constraints, and continuous variables for operation timing and memory dynamics.
Pipeline execution is optimized to minimize makespan under GPU memory limits including the GPU-CPU interconnect topology constraints. The system incorporates solver-side improvements (redundancy elimination, triangle inequality cuts, warm-start from AdaOffload, and cached solution reuse) and supports online schedule refinement. Experiments on up to 16 H100 GPUs and models up to 14.2B parameters show significant throughput improvements, especially in memory-limited settings.

**Strengths:**

- The MILP enforces dataflow correctness, single-resource exclusivity, F→B→W intra-stage semantics, activation-lifetime tracking, and PCIe-aware channel contention constraints. This captures more scheduling structure than prior heuristics.
- The formulation optimizes per-operation offload/reload placement using binary  W(i,j,c), rather than coarse strategies like all-F offload. It enables multi-objective trade-offs (memory vs. bubble elimination).
- Symmetry breaking, variable elimination, and triangle inequality cuts reduce the branch-and-bound search space theoretically. Warm-start with AdaOffload improves solver convergence.

**Weaknesses:**

- The paper does not provide statistics on: a) number of integer variables as a function of batch/stage counts, b) integrality gap evolution, c) memory footprint of solver state (if some it would be stored on the GPU side, despite the solver running on the CPU). This is critical because variable count grows as O(S⋅M⋅ops-per-microbatch).
- Limited baseline spectrum in memory-aware regime. All non-offloading baselines fail with OOM in low-memory settings. Comparisons therefore conflate feasibility with optimality. SPPO and SSDTrain (both cited) could be implemented as additional strong baselines.
-  All evaluation is based on fixed compute profiling from warm-up iterations. Runtime variability (NVLink / PCIe interference, GPU clock scaling) is not modeled nor stress-tested.
- The scheduling model presumes deterministic layer compute times and memory footprints; training regimes with dynamic sparsity or variable sequence lengths would violate this assumption.
- Important: online solver role insufficiently validated While “updates applied whenever an improved solution is discovered” is stated, no experiments quantify: convergence rate, schedule-switch overhead, and stability across hundreds of training steps.

**Questions:**

- What is the exact variable count and constraint count for the largest 16-GPU case?
- Given operation timing stochasticity, do MILP solutions become suboptimal mid-training? How often is re-solving triggered?
- Does the pipeline stall while switching schedules? If not, explain safe handoff mechanism.
- Can the formulation be extended to hybrid PP + TP + ZeRO with overlapping comm/compute?
- Does AdaOffload remain beneficial when memory budgets are not close to the feasibility boundary?

---

### Official Review · Reviewer_gTDx · 2025-10-30

**Soundness:** 3
**Presentation:** 2
**Contribution:** 2
**Rating:** 4
**Confidence:** 5

**Summary:**

This paper presents OptPipe, a mixed-integer linear programming (MILP)–based scheduler for pipeline parallelism (PP), designed to maximize throughput under memory constraints. The main contributions include a MILP formulation that jointly models memory usage and end-to-end makespan, as well as AdaOffload, an initialization strategy that improves the efficiency of the MILP solver. Empirical results demonstrate that OptPipe achieves over 30% higher throughput compared to existing heuristic-based methods. Further analysis indicates that the performance improvements primarily arise from more effective memory utilization, enabled by the flexibility of the MILP framework.

**Strengths:**

- Addresses an important problem: The paper tackles the crucial challenge of balancing activation memory consumption and throughput in pipeline parallelism (PP).
- Sound and well-formulated approach: The MILP-based problem formulation is clear, rigorous, and accounts for key practical constraints, including memory limits and computation–offload overlap.
- Strong empirical results: The experimental evaluation convincingly demonstrates the advantages of the MILP-based scheduler over predefined scheduling strategies, highlighting its flexibility and effectiveness.

**Weaknesses:**

- Novelty concern: The use of MILP for pipeline scheduling appears to have prior work. For example, Zero Bubble [1] employs an ILP-based solver with a similar action space—deciding binary precedence variables for forward, backward, and weight update passes. It would strengthen the paper to differentiate more clearly how OptPipe’s formulation or solver integration advances beyond these existing approaches.
- Scalability and computational cost: Since MILP is NP-hard, it is unclear how much additional optimization is achieved when the solver terminates due to a time limit. A quantitative analysis of MILP’s effectiveness under varying time budgets would be helpful. For instance, the authors could report how the trade-off between MILP runtime and achieved acceleration evolves, or compare the final speedup relative to the initialization-only schedule.
- Evaluation setup: Some of the experimental configurations appear unrealistic. For example, in Table 2, the tested model architecture (e.g., a 7B model with 256 layers and a hidden size of 128) deviates substantially from practical model designs. A more representative setup would make the evaluation results more convincing and relevant to real-world scenarios.
- Figure 2 closely resembles Figure 1 in [1], raising potential plagiarism or reuse concerns. The authors are encouraged to redraw or substantially modify this figure to ensure originality and avoid any misunderstanding.

Reference:
[1] Zero Bubble Pipeline Parallelism. https://arxiv.org/pdf/2401.10241

**Questions:**

- What is the key insight or improvement of the proposed MILP formulation compared to the ILP-based approach used in Zero Bubble?
- How are the execution times of the forward, backward, and weight update (F/B/W) passes determined? Are they obtained through profiling, estimation, or analytical modeling?
- How effective is the MILP optimization compared to the initialization scheme? In particular, is the initialization already sufficient to mitigate the limitations of PipeOffload-style scheduling, or does MILP provide a substantial additional gain?
- Do the empirical results generalize to practical model configurations (e.g., architectures similar to Qwen, LLaMA, or GPT-3)?

---

### Official Review · Reviewer_W9be · 2025-10-31

**Soundness:** 3
**Presentation:** 2
**Contribution:** 2
**Rating:** 4
**Confidence:** 4

**Summary:**

This paper presents OptPipe, a new framework for optimizing pipeline parallelism (PP) in LLM training, specifically addressing the trade-off between pipeline bubble latency and activation memory consumption. Unlike prior methods such as PipeOffload that rely on coarse-grained heuristics to manage activation offloading, OptPipe takes a principled optimization approach.

The core contribution is the formulation of the end-to-end pipeline scheduling problem—including all computation (Forward, Backward, Weight) and data transfer (Offload, Reload) operations —as a Mixed-Integer Linear Programming (MILP) model. The objective is to find a schedule that minimizes the total training makespan while strictly adhering to per-device memory constraints.

**Strengths:**

1. The primary strength is the shift from heuristic-based scheduling (like PipeOffload ) to a principled, formal optimization framework. Formulating the entire schedule, including offloading decisions, as an MILP  is a non-trivial and superior approach. It allows the system to find fine-grained, non-obvious schedules that heuristics would miss.

2. The "online scheduling" design (Figure 1) is a very clever and pragmatic solution to the NP-hard nature of MILP. By running the solver on the CPU asynchronously and dynamically updating the GPU schedule, the system gets the "best of both worlds": it starts training immediately with a good heuristic and converges toward an optimal schedule over time, hiding the solver's cost.

3. Table 1 robustly demonstrates OptPipe's superiority in the most critical, memory-limited scenarios, where it is >20% faster than the only other viable baseline, PipeOffload.

**Weaknesses:**

See Questions below.

**Questions:**

1. How critical is Gurobi to your results? Have you experimented with open-source MILP solvers? What is the performance degradation when using a solver like CBC or GLPK, both in terms of solver time and final schedule quality (throughput)? This is a key question for the practical impact on the open-source community.

2. Could you please provide an ablation that separates the gains from your AdaOffload heuristic and the MILP solver? Specifically, what is the throughput of just the AdaOffload schedule (the initial solution) compared to the final OptPipe schedule (the solution after Gurobi runs)? This would clarify how much benefit the complex MILP solving adds on top of your improved heuristic.

3. How long does the "Profile" phase take in your experiments (e.g., for the 14B model on 16 GPUs)? Furthermore, how often do you find the schedule needs to be re-optimized during a long training run if computation or communication times drift?

---

### Note · Authors · 2026-01-20

**Comment:**

We have decided to withdraw our paper from ICLR 2026. We thank the reviewers for their time and valuable feedback, particularly regarding the baselines and solver scalability. We will use these insights to improve the work for future presentation.

**Withdrawal Confirmation:**

I have read and agree with the venue's withdrawal policy on behalf of myself and my co-authors.